# Food Insecurity and Women’s Choice of Reversible Contraceptives: Differential Effects by Maternal Age

**DOI:** 10.3390/ijerph21101343

**Published:** 2024-10-10

**Authors:** Otobo I. Ujah, Pelumi Olaore, Russell S. Kirby

**Affiliations:** 1Department of Obstetrics and Gynecology, Federal University of Health Sciences, Otukpo 972261, Nigeria; otoboujah@yahoo.com; 2Chiles Center, College of Public Health, University of South Florida, Tampa, FL 33612, USA; pelumi@usf.edu

**Keywords:** contraception, Food Insecurity Experience Scale (FIES), material hardship, Multiple Indicator Cluster Survey (MICS), Nigeria

## Abstract

We investigated the relationships between food insecurity (FI) and women’s choice of reversible contraceptives, overall and according to the level of method effectiveness, among partnered women of reproductive age in Nigeria. This population-based cross-sectional analysis used nationally representative data from Round 6 of the UNICEF-supported Multiple Indicator Cluster Survey (MICS) conducted in Nigeria. The sample included married or in-union women aged 15–49 years who reported a live birth in the last 2 years preceding the survey (unweighted *N* = 8496). Survey-weighted multivariable binomial and multinomial logistic regression analyses were performed to generate estimates of the association between FI (none, moderate, and severe) and reversible contraceptive use (overall and by method effectiveness). A Bonferroni correction was used to account for multiple testing. We stratified the models by maternal age to describe the experiences of women aged 15–24 years, 25–34 years, and 35–49 years. Overall, 6438 (74.1%) of the women in the sample experienced food insecurity (moderate, *n* = 2559, 30.7%; severe, *n* = 3879, 43.4%). In the adjusted model, we observed no statistically significant association between experiencing MFI and SFI and the use of reversible contraceptives (overall and specific) after adjustment for multiple testing. The stratified analyses showed that among women aged 25–34 years in the sample, those experiencing SFI in the past 12 months, compared to their food-secure counterparts, had significantly lower odds of reporting the use of a least effective contraceptive method (OR, 0.53; 95% CI, 0.34–0.83; *p* = 0.0052). However, this failed to reach the significance threshold upon adjustment for multiple testing. We found no significant association between the FI levels and use of reversible contraceptives (overall and specific) among partnered women (15–49 years) in Nigeria who were 2 years postpartum following a live birth and who were at risk of pregnancy.

## 1. Introduction

Despite reported declines in the burden of unintended pregnancies globally, approximately 48% of all pregnancies were estimated to be unintended between 2015 and 2019 [1]. During this period, the rate of unintended pregnancy was highest in Sub-Saharan Africa (SSA) (91 per 1000 women aged 15–49 years) compared to other regions of the world [1]. Unintended pregnancy has serious health, social and economic implications for women and their families [2]. For example, about 13% of maternal deaths result from complications of unsafe abortion following the termination of unintended pregnancies [3]. Despite evidence of contraceptive use, overall and specific methods, being an effective strategy for preventing unintended pregnancies [4,5,6], the high burden of unintended pregnancy, especially among young and low-income women, reflects the persistence of barriers in accessing and utilizing effective methods of contraception [5,6,7].

Broadly, contraception can be classified as either reversible or permanent [8]. Reversible contraceptive methods allow women to temporarily delay or avoid pregnancy and comprise oral contraceptive pills, injectables, contraceptive patches, vaginal rings, contraceptive implants, barrier methods (such as condoms, diaphragms, cervical caps, and spermicides), and intrauterine devices (IUDs) [9]. Permanent methods, on the other hand, result in the permanent prevention of pregnancy and include tubal ligation for women and vasectomy for men [10,11,12].

Research shows that structural and economic factors may play an important role in shaping reproductive and contraceptive behaviors [7]. Yet, much remains to be understood about the specific ways in which the different dimensions of material hardship impact contraceptive use behaviors [13]. Material hardship occurs when individuals experience difficulties meeting basic daily food, housing, and transportation needs [6,14]. Food insecurity (FI), specifically, is a growing public health concern characterized by limited or uncertain access to safe and sufficient food, or the inability to acquire personally acceptable food through socially acceptable means in order to maintain a health and active lifestyle [15,16]. According to estimates from the United Nations Food and Agriculture Organization (UN FAO), in 2022, globally, approximately 2.3 billion people (29.6%) experienced moderate or severe FI in 2022, and 11.3% were severely food-insecure [17]. The prevalence of FI levels is particularly high in SSA, with estimates suggesting that at least two-thirds (67.2%) of the population were food-insecure in 2022 with nearly 27% experiencing severe FI [17]. Furthermore, women are disproportionately affected by FI compared to men [18,19,20].

While discussions of FI commonly center around its nutritional impact, such as malnutrition, there is growing interest in understanding the mechanisms through which FI may impact health behaviors and outcomes (i.e., non-nutritional consequences of FI) [21,22]. Studies have shown associations between FI and mental health, chronic diseases, and adverse pregnancy outcomes, including birth defects [23,24,25]. Various mechanisms have also been proposed to understand the mechanism between FI (or other forms of material hardships) and contraceptive behaviors. One such mechanism is based on the scarcity theory, which suggests that material hardship, including limited financial resources, constrains the cognitive bandwidth needed to make well-informed decisions, leading individuals to prioritize their most immediate necessities [7,26]. Consequently, this limitation impairs self-control, negatively impacting health behaviors such as contraceptive use [7]. Additionally, Field argues that material hardship can limit both the financial resources and cognitive abilities required for consistent contraceptive use and the long-term maintenance of these behaviors [6]. These limitations may be particularly pronounced for highly effective contraceptive methods, such as intrauterine devices (IUDs) and contraceptive implants, which can be costly and necessitate scheduled clinic appointments.

Another proposed mechanism for the influence of FI on contraceptive behavior is through its impact on mental health and maternal empowerment. Diamond-Smith et al. suggest that FI predicts mental health disorders, including anxiety and depression, which, consequently, affect women’s self-efficacy in accessing and advocating for essential services within their households or communities [27]. Furthermore, FI may contribute to a sense of disempowerment, further exacerbating the negative impact on contraceptive use [27].

While several observational studies have examined the relationship between FI and contraceptive behaviors [5,26,27], empirical evidence of this relationship in Nigeria, as in most parts of SSA, is limited. Nigeria is the most populous country in Sub-Saharan Africa, with an estimated population of over 200 million people in 2022. The country is divided into six geopolitical regions: North East, North West, North Central, South East, South West, and South South. Women of reproductive age (15–49 years) constitute approximately 20% of the population [28]. According to 2019 estimates, the total fertility rate in Nigeria is 5.3 live births per woman [29]. Despite this high fertility rate, only 40% of the demand for family planning is satisfied through modern contraceptive methods [30]. The modern contraceptive prevalence rate (mCPR) in Nigeria remains low, at approximately 12% [31]. According to estimates from the World Health Organization (WHO), Nigeria has an extremely high maternal mortality rate of 1047 deaths per 100,000 live births [32].

This study aims to build upon existing research to fill this important gap regarding evidence of the implications of FI for contraceptive and reproductive behaviors by investigating the associations between FI and women’s choice of reversible contraceptives, overall and according to the level of method effectiveness, among a nationally representative sample of partnered women of reproductive age (15–49 years) in Nigeria. We hypothesize that women experiencing moderate or severe FI will be less likely to use contraceptives overall and that these effects will vary by maternal age.

## 2. Materials and Methods

### 2.1. Study Design, Setting and Population

This cross-sectional study was based on a secondary analysis of nationally representative data derived from Round 6 of the MICS conducted in Nigeria. The Multiple Indicator Cluster Survey (MICS) is a cross-sectional survey conducted by the National Bureau of Statistics (NBS) in Nigeria, with support from UNICEF. The survey aims to collect data on sociodemographic and health indicators from households, children aged 0–5 years, women aged 15–49 years, and men aged 15–49 years. The survey utilizes a multistage stratified cluster sampling technique to ensure representative data collection. In the first stage, enumeration areas (EAs) were selected using a probability proportional to size (PPS) of the number of households, based on the 2006 Population and Housing Census of the Federal Republic of Nigeria (NPHC). In the second stage, 20 households were randomly selected from each EA. In households with more than one eligible respondent, only one respondent was randomly selected for interview. Data were collected using Computer-Assisted Personal Interviewing (CAPI) technology through face-to-face interviews with respondents in their respective households. The survey report provides more detailed information on the survey sampling design and data collection methods [33].

For this study, data on FI were obtained from the household file while data on contraceptive behavior were obtained from the women’s file. The analysis was restricted to women who were potentially at risk of pregnancy and included partnered (married or in union) women of reproductive age (15–49 years) who reported having at least one live birth within the two years prior to the survey. Observations were excluded for women who reported sterilization (either the female respondent or her current husband/partner), infertility, menopause, hysterectomy or being pregnant. Furthermore, individuals with missing data on FI, contraceptive behaviors, or covariates of interest were also excluded (Figure 1). As the sample was not self-weighting, the women’s sample survey weights included in the MICS dataset were used for analyzing and reporting results.

### 2.2. Outcomes

Our primary outcomes of interest were women’s overall contraceptive use and the effectiveness of the method of contraception women had chosen to use. These outcomes were ascertained based on women’s responses to two questions in the MICS. First, women were asked “Are you currently doing something or using any method to delay or avoid getting pregnant?” If participants responded affirmatively, they were then asked to specify the method(s) they were using, in response to the question “What methods are you using to delay or prevent pregnancy?”. Those using more than one method were classified according to the most effective method based on its typical use failure rate (<1%) [13,34].

Overall contraceptive use was modeled as a binary variable (any method/no method) while contraceptive method effectiveness was modeled as a 4-level outcome variable by grouping the methods according to their effectiveness and categorized as (1) most effective reversible methods: these include reversible contraceptive methods that last longer than three months, such as intrauterine devices (IUDs) and contraceptive implants; (2) moderately effective methods: injectables, pills, or diaphragms; (3) least effective methods: condoms (male and female), other barrier or traditional methods, emergency contraception, standard days method, and lactational amenorrhea method; and (4) no contraceptive method (nonuse).

### 2.3. Exposure

The main exposure variable was FI, assessed using the standardized eight-item Food Insecurity Experience Scale (FIES), which was developed by the United Nations Food and Agriculture Organization (UN FAO). The FIES serves as a tool for estimating the prevalence of moderate or severe food insecurity within populations, aligning with Sustainable Development Goal (SDG) indicator 2.1.2 [35]. This standardized scale allows for comparability across different studies and contexts. Participants were asked to recall their household’s experiences of food insecurity over the past 12 months and respond to each of the eight FIES questions with “Yes”, “No”, or “Don’t know” [33]. Raw household food insecurity scores, ranging from 0 to 8, were calculated as the sum of affirmative responses. Based on their scores, participants were categorized into three levels of food insecurity: none (scores 0–3), moderate (scores 4–6), or severe (scores 7–8), as in previous studies [35,36].

### 2.4. Control Variables

Several demographic, socioeconomic, and reproductive health variables related to food insecurity and contraceptive use were selected a priori for inclusion and adjusted for in the analyses based on a comprehensive literature review [5,6,7,20,27], their theoretical relevance and biological/mechanistic plausibility in the exposure–outcome relationship as well as the availability of the variables in the dataset. These covariates included maternal age (categorized as 15–24, 25–34, 35–49), religious affiliation (Christian vs. non-Christian), parity (low, average, and high), intendedness of last pregnancy (unplanned vs. planned), fertility intentions (wants more, wants none, undecided), health insurance coverage (yes vs. no), household wealth quintile (poorest, poorer, middle, richer, richest), place of residence (rural vs. urban), geographical region (North Central, North East, North West, South East, South South, and South West).

### 2.5. Statistical Analyses

All statistical analyses were performed using SAS version 9.4 (SAS Institute, Cary, NC, USA) and plots presented were generated using the R software version 4.3.2. To account for the complex survey design, such as stratified sampling and probabilities of unequal sample selection between clusters, all analyses were survey-weighted by using the national women’s weighting for the entire MICS to generate nationally representative estimates.

Descriptive statistics were calculated using the SAS SURVEYMEANS and SURVEYFREQ procedures to estimate weighted means (SE) and frequencies (%), respectively, for women’s characteristics and contraceptive use. Furthermore, the Rao–Scott χ^2^ test was used to examine differences in the proportion of women reporting FI across various subgroups of categorical variables. Weighted multivariable analyses were conducted to assess the association between each contraceptive behavior outcome with the levels of FI. Separate binomial and multinomial logistic regression models were fitted using the SURVEYLOGISTIC procedures, with the logit and glogit functions, respectively. All models were adjusted for the aforementioned covariates. To evaluate the potential influence of age, we stratified the models by maternal age group, operationalized as a polytomous variable categorized as 15–24 years, 25–34 years, and 35–49 years. All tests were two-tailed. Given that multiple comparisons were performed in this study, we adjusted for multiple testing with *p* values adjusted using the Bonferroni–Holm correction method, with adjusted *p* values considered statistically significant only if <0.05/n (where n is the number of tests conducted).

## 3. Results

### 3.1. Characteristics of the Sample

The weighted sample included 8362 participants residing in 1672 communities. Of these, 6196 (74.1%) of the participants lived in food-insecure households, with 2559 (30.7%) experiencing MFI and 3879 (43.4%) experiencing SFI. The mean (SE) age of the participants was 29.5 (0.12) years. Table 1 presents the characteristics of the participants overall and according to the food insecurity status. Most of the participants (48.3%) were in the 25–34 years age group. Of the survey participants, more than three-quarters (78.3%) reported their most recent pregnancy as planned, about 72% expressed desire for further childbearing and 62% were affiliated with religions other than Christianity. Furthermore, only about 3% of the survey respondents had health insurance coverage and nearly one in two (47.8%) resided in poor households. About 64% of the respondents were residing in rural areas and 64% were living in the Northern region. The results of the univariate analyses revealed that the maternal age, parity, intendedness of last pregnancy, health insurance coverage household wealth quintile, place of residence, and geographical region were significantly associated with experiences of FI (all *p* < 0.05). Fertility intentions and religious affiliation were not associated with experiences of FI. Figure 2 shows the pattern of contraceptive method use across the computed FIES scores. The median (IQR) FIES score was 6 (3). Among those with a median FIES score (*n* = 1071, 12.2%), 814 (80%) were not using a contraceptive method, 91 (8.9%) were using a least effective method, 73 (7.2%) were using a moderately effective method, while 39 (3.8%) were using a most effective method.

As shown in Table 2, there were no significant differences in the prevalence of any contraceptive use based on the FI status. Overall, 1554 (20.4%) women reported they were using a contraceptive method. However, the prevalence was slightly higher among those who were food-secure (*n* = 376, 22.2%) and moderately food-insecure (*n* = 470, 21.8%) (*p* = 0.053). Among those using a contraceptive method, 589 (8.1%) were using a less effective method, 617 (7.6%) were using a moderately effective method, while 348 (4.7%) were using a most effective method.

Overall, across all the age groups, more than three-quarters of the participants reported not using a method of contraception. The use of most effective contraceptive methods was lowest across all the age categories and was higher with increasing age (15–24 years, 3.52%; 24–35 years, 4.9%; 35–49 years, 5.2%). The use of a least effective method was highest among the participants aged 15–24 years (6.2%) and 25–34 years (9.0%), while the use of a moderately effective method was highest for those aged 35–49 years (9.3%). Figure 3 shows the distribution of the bivariate relationships between the FI category and the choice of reversible contraceptive methods across the different age groups. The results of the stratified analyses highlight substantial variations in contraceptive use across the different age groups and levels of FI. Among the participants in the 25–34 years age group, there was a significant relationship between FI and contraceptive method use (*p* = 0.0007). However, no significant relationships were observed for those in the 15–24 year and 35–49 years age groups (*p* > 0.05).

### 3.2. Association between Food Insecurity and Contraceptive Use

As shown in Table 3, we observed no significant relationship between experiencing MFI and overall contraceptive use in both the unadjusted (OR: 0.98; 95% CI: 0.77–1.24) and adjusted (OR: 1.17; 95% CI: 0.92–1.48) binomial models. Similarly, although there was a negative association between experiencing SFI and the overall contraceptive use, compared with no method use, this association was not significant in both the unadjusted (OR: 0.79; 95% CI: 0.62–1.01) and adjusted (OR: 0.97; 95% CI: 0.76–1.23) binomial models.

In the multinomial logistic regression analyses, the women experiencing MFI and SFI had lower odds of current use of least effective methods in both the crude and adjusted models. However, these associations were not statistically significant. Similarly, the women experiencing MFI and SFI had higher odds of current use of moderately effective methods in both the crude and adjusted models; these associations did not reach the significance threshold even after adjustment for multiple testing. No associations were observed between the experiences of MFI and SFI and current use of most effective methods either in the crude or adjusted models.

After stratifying the analyses by maternal age groups, women experiencing MFI were no more or less likely to report overall and method-specific contraceptive use, compared to their counterparts who were food-secure, in those aged 15–24 years, 25–34 years, and 35–49 years (Table 4). In the women in the sample aged 25–34 years, those experiencing SFI, compared to their food-secure counterparts, had significantly lower odds of reporting use of least effective contraceptive methods (OR, 0.53; 95% CI, 0.34–0.83; *p* = 0.0052). However, this failed to reach the statistical significance threshold upon correction for multiple testing. For all the other categories, those experiencing SFI were no more or less likely to report overall and method-specific contraceptive use, compared to their counterparts who were food-secure, among women aged 15–24 years, 25–34 years, and 35–49 years.

## 4. Discussion

In this study, including a nationally representative sample of partnered women of reproductive age in Nigeria who had a live birth in the past 2 years and who were at risk of pregnancy, approximately three out of four of the participants experienced MFI or SFI. Only about one in five of the women were using a reversible method of contraception, with most using either a least or moderately effective method. Contrary to our hypotheses, we did not find clear evidence of a statistically significant association between experiencing MFI and SFI in the past 12 months and the use of reversible contraceptive methods (overall and specific) among partnered women with a live birth in the past 2 years who were at risk of pregnancy. Stratification by maternal age did not result in an appreciable change in the findings after adjusting for multiple testing, although a possible negative association between SFI and the use of least effective methods was observed among women 25–34 years prior to adjusting for multiple testing.

Despite evidence from limited observational studies suggesting that the contraceptive and reproductive behaviors of women are negatively affected by several dimensions of material hardship, including FI, the results of this study, nonetheless, do not provide conclusive evidence to support this association. Our findings of a null association between FI and reversible contraceptive use differ from those of previously published studies which found a positive association. The magnitude of our observed association for MFI and SFI is somewhat comparable to the results of a population-based cross-sectional analysis among women in Nepal, suggesting an inverse, but not statistically significant, relationship between family planning use and increasing levels of FI (MFI: aOR = 0.92, 95% CI 0.77–1.10; SFI: a0.77, 95% CI = 0.63–0.94). However, the study did not disaggregate the effects of FI according to the levels of effectiveness of the contraceptive method use [27]. Another study, which included 651 married women of reproductive age in Ethiopia [5], reported an odds ratio of 1.69 (95%, 1.03–2.66) for the association between FI and modern contraceptive use. In a cross-sectional study in the United States (US), Lin et al. [37] reported an odds ratio of 1.9 (95% CI: 1.1–3.2) for the association between food, transportation, and/or housing insecurity and difficulty accessing contraceptives during the COVID-19 pandemic. Interestingly, this study used the FIES to measure FI. This discrepancy points to the possible influence of other contextual factors, along with the combined effects of different dimensions of material hardships on the dynamics of contraceptive behavior. Moreover, Field [6], demonstrated that women experiencing one form of material hardship were not more or less likely to use contraception, compared to those without material hardship.

There are several plausible explanations for the differences between our findings and previous findings. The first relates to the composition of our study sample. Previous studies evaluated contraceptive use, regardless of the method type, either among the general population of women, young women [6], or current users of contraception [13], and also did not account for women who were at risk of pregnancy and for who the unmet need for contraception is greatest. Furthermore, while we were able to control for several confounders that were identified based on existing evidence and biologic plausibility, we were unable to account for other significant covariates, including maternal education, occupation, and partner-related factors. These unaccounted factors could have potentially influenced the findings of our study and the conclusions drawn thereof. Moreso, the likelihood of residual confounding could potentially explain why our findings are inconsistent with the previous literature. Additionally, the different results may also be due to differences in the measurement of FI. While existing studies on FI have previously relied on self-reports, as with ours, to our knowledge, our study used a scale which has been validated in a variety of populations [35,38]. Consequently, further research is warranted to ascertain whether FI, as assessed using validated measures, is indeed associated with contraceptive behaviors among women who have an unmet need for contraception. Although our primary focus was on FI, other forms of material hardship, such as an inability to afford transportation costs to health facilities—even when user fees for contraceptives are waived—may also hinder the use of effective contraceptive methods among women. For instance, a previous study involving women living with HIV in Tanzania found that higher transportation costs were linked to a lower likelihood of using modern contraceptive methods [39].

Given the growing interest in understanding the negative consequences of FI on health behaviors and outcomes, this study makes substantial contributions to the existing literature in several ways. First, we assessed the influence of FI among women who were within 2 years postpartum for who the need for contraceptive uptake remains critical in order to ensure an optimal interpregnancy interval by avoiding unintended and rapid repeat pregnancies. Indeed, evidence suggests that the unmet need for contraception is greatest within the first 2 years after childbirth [40]. Additionally, we explored the differential effects of FI (none, moderate, and severe) rather than simplifying FI as secure and insecure, which has the potential to mask disparities in the effects of FI. Moreover, FI was assessed using an internationally validated scale which allows for the comparability of findings across different studies and across different contexts.

The current findings should be interpreted in the light of several considerations. The cross-sectional nature of our data limits our ability to draw causal assumptions or capture temporal trends in the relationship between food insecurity and contraceptive use. Additionally, the reliance on self-reported measures of the exposure and outcomes increases the likelihood of recall bias, as individuals may have different interpretations or recollections of their experiences. Lastly, our study focuses on contraceptive dynamics on a subsample of married/in-union women of reproductive age who reported a live birth in the 2 years preceding the survey. As a result, the generalizability of our findings may be limited in terms of reflecting the broader population of women.

While contraceptive utilization is an important consideration in understanding contraceptive behavior, future research should also focus on investigating the potential association between food insecurity (FI) and the accessibility and affordability of contraception. Emerging evidence suggests that FI is linked to delays in obtaining necessary maternal preventive and medical care, as well as the potential abandonment of such care. Therefore, examining this relationship across various settings would provide valuable insights for informing policy and interventions. Also, longitudinal study designs would be beneficial in establishing causal relationships and identifying temporal trends over time. Furthermore, employing mixed-methods approaches would provide a more nuanced perspective by considering the various influences that impact contraceptive behavior among women experiencing different levels of food insecurity. In terms of implications for practice, there is a growing enthusiasm for consistent screening and measurement of FI in healthcare settings [41,42]. By offering personalized support, women’s and reproductive health providers can play an important role by pragmatically identifying women who are food-insecure or at risk and linking them with community-based food support programs, thereby mitigating the adverse effects of health and well-being [42]. However, the evidence of the extent to which this can be implemented in developing settings remains unclear.

## 5. Conclusions

The present findings from this cross-sectional analysis among a nationally representative sample of partnered women of reproductive age in Nigeria with a live birth in the past 2 years and who were at risk of pregnancy suggests that experiencing MFI and SFI in the past 12 months is not associated with overall and method-specific use of reversible contraceptives after accounting for multiple testing. Further, we found no evidence of heterogeneity in the associations between the experiences of FI and the women’s choice of reversible contraceptives by maternal age.

## Figures and Tables

**Figure 1 ijerph-21-01343-f001:**
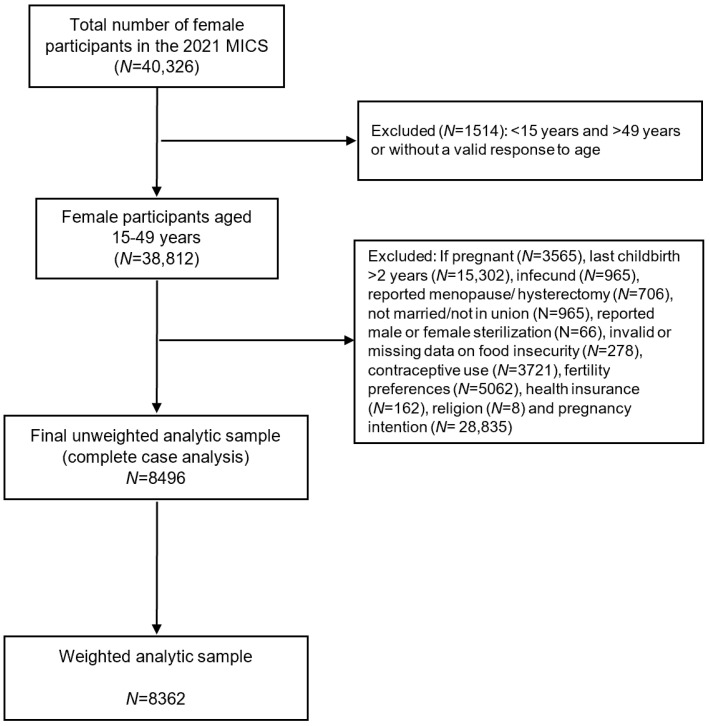
Flow chart of participants in the 2021 MICS who were included in the analysis. Note: since women can display more than one of these exclusion criteria, the sum of the subsets was greater than the total excluded. The weighted size of the analytic sample was derived by adjusting for appropriate sample weight, stratification, and clustering in order to account for the complex survey design as recommended in the methodology of the MICS.

**Figure 2 ijerph-21-01343-f002:**
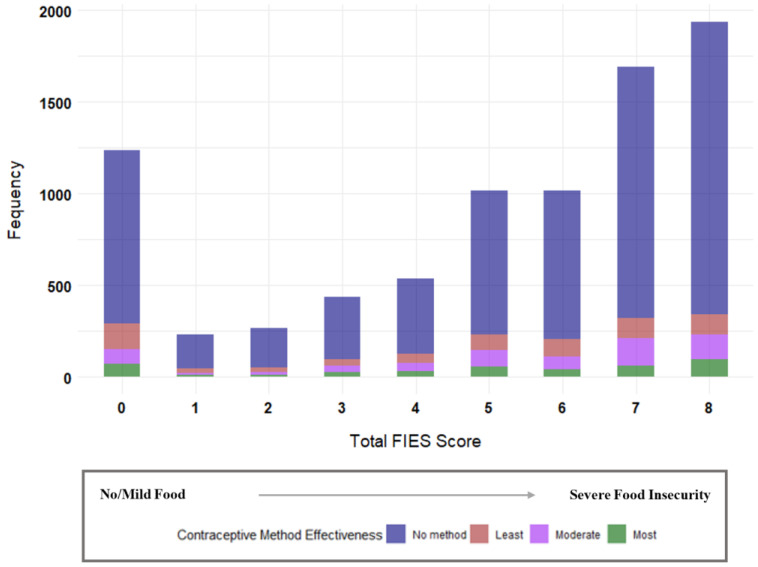
Total Food Insecurity Experience Scale (FIES) score and contraceptive method use.

**Figure 3 ijerph-21-01343-f003:**
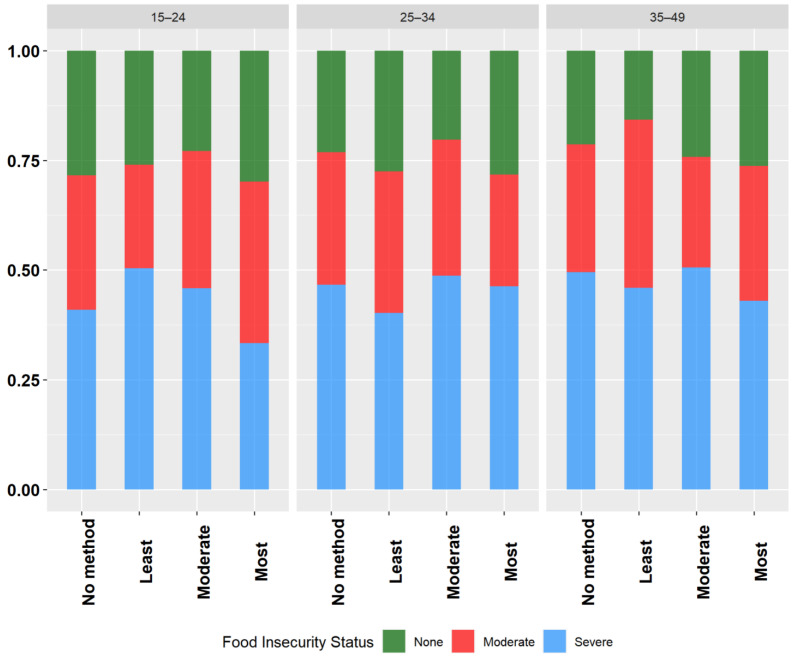
Weighted proportions showing the relationships between women’s choice of reversible contraceptive method and food insecurity category according to age groups, Nigeria, MICS, 2021.

**Table 1 ijerph-21-01343-t001:** Characteristics of partnered women (15–49 years), overall and according to food insecurity levels, with a live birth in the past 2 years and at risk of pregnancy, Nigeria, Multiple Indicator Cluster Survey, 2021.

			Food Insecurity									
Participant Characteristics	Total Sample		None			Moderate			Severe			
	*N*	%	*n*	%	95% CI	*n*	%	95% CI	*n*	%	95% CI	*p* Value
**Overall**	8496	100	2058	25.91	24.24–27.58	2559	30.70	29.24–32.16	3879	43.39	41.66–45.11	**<0.0001**
**Age**												
15–24	2276	25.34	638	27.26	24.62–29.90	693	25.43	23.11–27.74	945	24.13	22.18–26.06	**0.0162**
25–34	4113	48.34	967	50.10	47.02–53.18	1244	47.84	45.31–50.38	1902	47.84	45.31–50.38	
35–49	2107	26.32	453	22.64	19.81–25.46	622	26.73	24.32–29.14	1032	28.24	26.28–30.22	
**Parity**												
Low (1–2)	2765	34.14	817	41.64	38.69–44.59	836	33.15	30.46–35.85	1112	30.36	28.31–32.41	**<0.0001**
Average (3–4)	2606	30.92	610	30.50	27.85–33.16	772	31.41	28.79–34.04	1224	30.81	28.86–32.76	
High (5+)	3125	34.94	631	27.86	24.94–30.78	951	35.44	32.82–38.05	1543	38.83	36.65–41.01	
**Intendedness of last pregnancy**												
Unplanned	1791	21.63	1652	79.69	76.77–82.63	2069	81.55	79.44–83.66	2984	75.33	73.37–77.29	**0.0002**
Planned	6705	78.37	406	20.30	17.37–23.23	490	18.45	16.34–20.56	895	24.67	22.70–26.63	
**Fertility intentions**												
Wants none	1610	20.62	349	20.63	17.53–23.72	456	18.54	16.07–20.99	805	20.09	20.08–24.09	0.257
Wants more	6167	71.53	1538	71.86	68.74–74.99	1887	73.51	70.74–76.27	2742	69.94	67.55–72.33	
Undecided	719	7.85	171	7.51	5.81–9.22	216	7.99	6.41–9.5	332	7.97	6.63–9.31	
**Religious affiliation**												
Christian	3107	37.99	653	37.79	33.50–42.07	910	36.77	33.29–40.25	1544	38.98	35.91–42.06	0.586
Non-Christian	5389	62.01	1405	62.21	57.93–66.50	1649	63.23	59.75–66.71	2335	61.02	57.94–64.09	
**Health insurance coverage**												
No	8312	98.11	1968	94.59	92.92–96.26	2514	97.28	95.85–98.72	3830	98.49	97.70–99.28	**0.0001**
Yes	184	2.89	90	5.41	3.75–7.08	45	2.72	1.28–4.15	49	1.51	0.72–2.29	
**Household Wealth quintile**												
1 (Poorest)	2435	24.50	469	19.14	16.07–22.21	727	24.76	21.89–27.64	1239	27.51	24.91–30.11	**<0.0001**
2	2174	23.29	498	20.94	18.10–23.78	687	24.38	21.54–27.22	989	23.92	21.57–26.27	
3	1684	18.17	353	14.97	12.69–17.25	385	17.52	15.30–19.73	561	20.54	18.38–22.70	
4	1285	16.86	339	16.29	13.65–18.93	385	16.96	14.56–19.35	561	17.13	14.92–19.34	
5 (Richest)	918	17.19	399	28.67	24.10–33.24	253	16.39	13.57–19.20	266	10.91	9.04–12.78	
**Place of residence**												
Urban	2229	35.95	634	41.37	36.33–46.41	641	34.44	30.41–38.48	954	33.77	30.02–37.52	**0.0023**
Rural	6267	64.05	1424	58.63	53.59–63.67	1918	65.56	61.52–69.60	2925	66.23	62.48–69.98	
**Geographical region**												
North Central	1627	13.95	333	10.81	8.83–12.79	484	13.81	12.02–15.60	819	15.92	14.08–17.78	**0.0052**
North East	1953	15.74	544	16.81	14.03–19.59	531	14.26	11.99–16.54	878	16.15	13.91–18.39	
North West	2567	34.44	660	34.44	30.75–38.12	836	36.62	33.39–39.85	1071	32.90	30.05–35.76	
South East	726	7.93	105	6.97	3.01–10.93	220	7.80	5.22–10.39	401	8.60	6.48–10.72	
South South	780	10.17	143	8.88	6.61–11.15	253	10.23	8.30–12.15	384	10.91	9.36–1246	
South West	843	17.76	273	22.10	18.64–25.55	235	17.27	14.56–19.98	335	15.51	13.34–17.68	

Notes: All percentages (%) are column percentages and were calculated based on the weighted sample (*N* = 8362). The numbers of participants (*n*) in each category were computed based on the unweighted sample (*N* = 8496). *p* values were calculated using Rao–Scott chi-square tests. Abbreviations: CI—confidence interval.

**Table 2 ijerph-21-01343-t002:** Prevalence of Reversible Contraceptive Choices among Partnered Women with a Live Birth in the Past 2 Years at Risk of Pregnancy, Overall and by Food Insecurity Status: Nigeria, MICS, 2021.

	Overall		No FI		Moderate FI		Severe FI		
	*N*	%	*N*	%	*N*	%	*N*	%	*p* Value
Total	8496	100	2058	100	2559	100	3879	100	
**Overall contraceptive use**									
No	6942	79.62	1682	77.85	2089	78.19	3171	81.70	0.0525
Yes	1554	20.38	376	22.15	470	21.81	708	18.30	
**Type of contraceptive method use**									
None	6942	79.2	1682	77.85	2089	78.19	3171	81.70	**0.0189**
Least effective method	589	8.06	143	10.33	188	8.87	258	6.14	
Moderately effective method	617	7.63	136	6.43	181	8.27	300	7.90	
Most effective method	348	4.68	97	5.38	101	4.67	150	4.26	

Notes: All percentages (%) are column percentages and were calculated based on the weighted sample (*N* = 8362). The numbers of participants (*n*) in each category were computed based on the unweighted sample (*N* = 8496). *p* values were calculated using Rao-Scott Chi-square tests. Abbreviations: FI—Food Insecurity.

**Table 3 ijerph-21-01343-t003:** Associations between food insecurity status and the odds of contraceptive use, overall and by specific method (weighted *N* = 8362).

	Crude Estimates				Adjusted Estimates			
	Moderate FI		Severe FI		Moderate FI		Severe FI	
	OR (95% CI)	*p*-adj	OR (95% CI)	*p*-adj	OR (95% CI)	*p*-adj	OR (95% CI)	*p*-adj
**Contraceptive outcomes**								
**Overall contraceptive use ^†^**								
No (reference category)	1.00		1.00		1.00		1.00	
Yes	0.98 (0.77–1.24)	0.8699	0.79 (0.62–1.01)	0.0581	1.17 (0.92–1.48)	0.2029	0.97 (0.76–1.23)	0.7783
**By contraceptive method effectiveness ^‡^**								
No method (reference category)	1.00		1.00		1.00		1.00	
Least effective method	0.85 (0.57–1.27)	0.4397	0.57 (0.37–0.87)	0.0089	0.99 (0.69–1.44)	0.9774	0.69 (0.48–1.01)	0.0549
Moderately effective method	1.28 (0.89–1.84)	0.1803	1.17 (0.85–1.62)	0.3393	1.57 (1.06–2.32)	0.0251	1.43 (0.99–2.06)	0.0530
Most effective method	0.86 (0.56–0.87)	0.5152	0.75 (0.51–1.12)	0.1578	1.05 (0.67–1.63)	0.8421	0.94 (0.63–1.39)	0.7494

Notes: All estimates are weighted using women’s sampling weights included in the Multiple Indicator Cluster Survey (MICS) to account for stratification and clustering in the survey. The analytic sample includes partnered women (married or in union) of reproductive age with at least one live birth in the 2 years preceding the survey. ^†^ Binomial logistic regression (estimates generated using PROCSURVEY LOGISTIC with logit link function). ^‡^ Multinomial logistic regression (estimates generated using PROCSURVEY LOGISTIC with glogit link function). Abbreviations: AOR = adjusted odds ratio, CI = confidence interval, *p*-adj = *p* values adjusted for multiple hypotheses testing using the Holm–Bonferroni method.

**Table 4 ijerph-21-01343-t004:** Adjusted models (binomial and multinomial) of reversible contraceptive method use by food insecurity levels, stratified by maternal age (weighted *N* = 8362).

		Any Reversible Method vs. No Method ^†^		Least Effective vs. No Method		Moderately Effective vs. No Method		Most Effective vs. No Method	
Variable	Age, y	AOR (95% CI)	*p*-adj	AOR (95% CI)	*p*-adj	AOR (95% CI)	*p*-adj	AOR (95% CI)	*p*-adj
**Food insecurity category**									
Moderate food insecurity	15–24	1.27 (0.78–2.08)	0.3417	0.99 (0.46–2.13)	0.9753	1.49 (0.75–2.95)	0.2540	1.49 (0.54–4.11)	0.4389
(Reference = food-secure)	25–34	1.02 (0.72–1.43)	0.9329	0.79 (0.49–1.25)	0.3131	1.64 (0.90–2.98)	0.1043	0.90 (0.50–1.62)	0.7344
	35–49	1.38 (0.89–2.12)	0.1469	1.47 (0.70–3.09)	0.3046	1.53 (0.78–3.02)	0.2180	1.08 (0.57–2.08)	0.8070
Severe food insecurity									
(Reference = food-secure)	15–24	1.17 (0.74–1.85)	0.5159	0.89 (0.43–1.82)	0.7385	1.32 (0.69–2.48)	0.3954	1.44 (0.61–3.37)	0.4065
	25–34	0.89 (0.66–1.22)	0.4814	0.53 (0.34–0.83)	0.0052	1.64 (0.96–2.79)	0.0711	0.99 (0.60–1.64)	0.9887
	35–49	0.99 (0.66–1.48)	0.9438	0.94 (0.44–1.98)	0.8613	1.29 (0.72–2.31)	0.4021	0.69 (0.37–1.28)	0.2376

Notes: All estimates are weighted using women’s sampling weights included in the Multiple Indicator Cluster Survey (MICS) to account for stratification and clustering in the survey. The analytic sample includes partnered women (married or in union) of reproductive age with at least one live birth in the 2 years preceding the survey. ^†^ Binomial logistic regression (estimates generated using PROCSURVEY LOGISTIC with logit link function). Multinomial logistic regression (estimates generated using PROCSURVEY LOGISTIC with glogit link function). Abbreviations: OR = odds ratio, CI = confidence interval, *p*-adj = *p* values adjusted for multiple hypotheses testing using the Holm–Bonferroni method.

## Data Availability

The anonymized data that support the findings of this study are publicly available and can be accessed from https://mics.unicef.org/surveys (accessed on 17 January 2024).

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
