# Peer review of "Food Insecurity and Women’s Choice of Reversible Contraceptives: Differential Effects by Maternal Age"

_ijerph, 2024, doi:10.3390/ijerph21101343_

Round 1
Reviewer 1 Report
Comments and Suggestions for Authors
Thank you for allowing me to participate in this review and for studying such interesting topics as this one.
First of all, congratulations on an excellent introduction to the topic, although I missed the details of reversible contraceptive methods, defining them or detailing the existing types.
Continuing with the methodology, the sampling carried out was very well detailed.
The representation of the results and discussion of the results is correct.
Congratulations on the work carried out.
Author Response
Comments and Suggestions for Authors: First of all, congratulations on an excellent introduction to the topic, although I missed the details of reversible contraceptive methods, defining them or detailing the existing types.
Author’s response: We now detail the existing types of reversible contraceptive methods in the second paragraph of the introduction section
Reviewer 2 Report
Comments and Suggestions for Authors
Article with scientific and thematic relevance, of interest to different disciplinary areas, introducing current topics, with important intersections between food insecurity issues and the impact on potential reproductive and health outcomes.
The article has solid content, a good structure and is well written
The introduction sets out the relevance of the topic and the objectives of the work presented.
Key concepts for the study are introduced, such as food insecurity, which stems from a broader concept of material hardship.
The aim of the work, the target audience and the source of information are presented in a clear and well-articulated manner.
Following on from the results achieved, there is a good discussion and reflection on the possible reasons for the divergence between the results of this study and previous ones.
In this sense, the article seems to me to be very pertinent and worthy of publication.
I would, however, suggest a few improvements:
- The introduction lacks background information on the country, particularly in social and demographic terms, as well as reproductive health indicators.
- In the presentation of the objectives and the explanatory variables considered, it is important to justify the choice of the variables selected in a more robust way and also the greater investment in some of them (apart from age) in the presentation of the results, including the scales selected.
Author Response
Comments and Suggestions for Authors:
- The introduction lacks background information on the country, particularly in social and demographic terms, as well as reproductive health indicators
Author’s response: We made changes according to this suggestion in the fifth paragraph of the introduction section.
- In the presentation of the objectives and the explanatory variables considered, it is important to justify the choice of the variables selected in a more robust way and also the greater investment in some of them (apart from age) in the presentation of the results, including the scales selected.
Author’s response: We made changes according to this suggestion.
Reviewer 3 Report
Comments and Suggestions for Authors
The unintended pregnancies are still a health concern worldwide including in In sub-Saharan Africa where the prevalence is the highest in the world. The main cause of unintended pregnancies is the non-use or inconsistent use of contraceptive methods, due to a number of factors including structural and economic factors. Building on previous studies investigating the association between material hardship and reproductive behaviour, particularly contraceptive behaviour, this article investigates the relationship between food insecurity and reversal contraceptive use in Nigeria. The study hypothesises the mechanism under with food insecurity might influence contraceptive use on the scarcity theory, which suggests that material hardship constrains the cognitive bandwidth needed to make well-informed decisions including that of using contraceptive methods. Come of highly effective contraception may be costly and require health professional to insert or administer.
The article uses MICS data from Nigeria and applies binary and multinominal logistic regression the investigate weather severe and moderate food insecurity affects contraceptive behaviour as measured by overall contraceptive use and the effectiveness of the chosen contraceptive method. The food insecurity measures were derived using the standardized eight-item Food Insecurity Experience Scale developed by FAO. Contrary to the previous studies, this study did not find any significant relationship between both moderate and severe food insecurity and contraceptive use. The authors explain the difference in based on difference in sample composition, the fact that some important covariates were not controlled for in the previous studies among other factors.
However, if some of the important mechanism of which food insecurity may influence contraceptive uses has to do with the costly of getting contraception, it would also be important to know if the fact that in some country contraception is free of charge (not sure about Nigeria), although there might be some costs, like transport cost to the health facility, could also affect the relationship under investigation. Also, I am wondering whether an multilevel modelling wouldn’t be the best approach to this study.
Otherwise the study is well written and very nice to read.
Author Response
Comments and Suggestions for Authors:
However, if some of the important mechanism of which food insecurity may influence contraceptive uses has to do with the costly of getting contraception, it would also be important to know if the fact that in some country contraception is free of charge (not sure about Nigeria), although there might be some costs, like transport cost to the health facility, could also affect the relationship under investigation.
Author’s response: We made changes according to this suggestion.
- Also, I am wondering whether an multilevel modelling wouldn’t be the best approach to this study.
Author’s response: We chose not to use multilevel modelling as our primary objective was not to differentiate between compositional and contextual effects. However, exploring contextual effects to understand how women’s contraceptive behavior dynamics vary across communities in the context of food insecurity would be an interesting area for future research.